# A genome-wide meta-analysis identifies 50 genetic loci associated with carpal tunnel syndrome

Astros Th. Skuladottir [1✉], Gyda Bjornsdottir [1], Egil Ferkingstad [1], Gudmundur Einarsson [1], Lilja Stefansdottir[1], Muhammad Sulaman Nawaz [1,2], Asmundur Oddsson [1], Thorunn A. Olafsdottir [1], Saedis Saevarsdottir [1,2,3], G. Bragi Walters [1,2], Sigurdur H. Magnusson [1], Anna Bjornsdottir[4], Olafur A. Sveinsson[2], Arnor Vikingsson[3], Thomas Folkmann Hansen [5,6], Rikke Louise Jacobsen[7], Christian Erikstrup [8], Michael Schwinn[7], Søren Brunak [6], Karina Banasik [6], Sisse Rye Ostrowski [7,9], Anders Troelsen [10], Cecilie Henkel [11], Ole Birger Pedersen [12✉], DBDS Genetic Consortium*, Ingileif Jonsdottir [1,2], Daniel F. Gudbjartsson [1,13], Patrick Sulem [1], Thorgeir E. Thorgeirsson[1], Hreinn Stefansson[1] & Kari Stefansson [1,2✉]

Carpal tunnel syndrome (CTS) is the most common entrapment neuropathy and has a largely unknown underlying biology. In a genome-wide association study of CTS (48,843 cases and 1,190,837 controls), we found 53 sequence variants at 50 loci associated with the syndrome. The most significant association is with a missense variant (p.Glu366Lys) in *SERPINA1* that protects against CTS ($P = 2.9 \times 10^{-24}$, OR = 0.76). Through various functional analyses, we conclude that at least 22 genes mediate CTS risk and highlight the role of 19 CTS variants in the biology of the extracellular matrix. We show that the genetic component to the risk is higher in bilateral/recurrent/persistent cases than nonrecurrent/nonpersistent cases. Anthropometric traits including height and BMI are genetically correlated with CTS, in addition to early hormonal-replacement therapy, osteoarthritis, and restlessness. Our findings suggest that the components of the extracellular matrix play a key role in the pathogenesis of CTS.

[1] deCODE genetics/Amgen Inc., Reykjavik, Iceland. [2] Faculty of Medicine, University of Iceland, Reykjavik, Iceland. [3] Landspitali—the National University Hospital of Iceland, Reykjavik, Iceland. [4] Heilsuklasinn Clinic, Reykjavik, Iceland. [5] Danish Headache Center, Department of Neurology, Copenhagen University Hospital, Rigshospitalet—Glostrup, Glostrup, Denmark. [6] Novo Nordisk Foundation Center for Protein Research, Faculty of Health and Medical Sciences, University of Copenhagen, Copenhagen, Denmark. [7] Department of Clinical Immunology, Copenhagen University Hospital—Rigshospitalet, Copenhagen, Denmark. [8] Department of Clinical Immunology, Aarhus University Hospital, Aarhus, Denmark. [9] Department of Clinical Medicine, Faculty of Health and Medical Sciences, University of Copenhagen, Copenhagen, Denmark. [10] Department of Orthopaedic Surgery, CAG ROAD - Research OsteoArthritis Denmark, Copenhagen University Hospital, Hvidovre, Denmark. [11] Department of Orthopaedic Surgery, CORH, Copenhagen University Hospital, Hvidovre, Denmark. [12] Department of Clinical Immunology, Zealand University Hospital—Køge, Køge, Denmark. [13] School of Engineering and Natural Sciences, University of Iceland, Reykjavik, Iceland. *A list of authors and their affiliations appears at the end of the paper.
✉email: astros.skuladottir@decode.is; olbp@regionsjaelland.dk; kstefans@decode.is

E ntrapment neuropathies are common and debilitating. In these neuropathies, a peripheral nerve becomes compressed where it passes through narrow anatomical areas, due to swelling of surrounding tissues or anatomical abnormalities[1]. Carpal tunnel syndrome (CTS) is the most common form of entrapment neuropathy, affecting around 3% of women and 2% of men at some stage in their lives[2]. It is caused by elevated pressure on the median nerve in the carpal tunnel[3], a space-limited osteofibrous canal that, in addition to the median nerve, is bounded by the carpal bones, the transverse carpal ligament, and the flexor tendons[4]. The median nerve innervates the thenar muscles and the radial lumbricals, and CTS patients commonly experience severe pain and paresthesia, and less commonly weakness, in the median nerve distribution[4].

Although CTS has been widely studied, its pathophysiology is not fully understood. A combination of factors such as mechanical trauma (e.g. wrist fractures), fibrosis, and inflammation of tendons can lead to pressure on the median nerve within the carpal tunnel[5]. Other risk factors include age, female sex[2], pregnancy[6], obesity, diabetes, and heredity[7].

The diagnosis of CTS is, together with medical history, obtained through medical assessment such as Phalen's and Tinel's test, and/or electrophysiological measurement[4]. Several treatments are available for the management of CTS, ranging from lifestyle changes (e.g. reduction of certain work activities) to local corticosteroid injection, and in the most severe cases, surgical carpal tunnel decompression.

The estimated familial occurrence of CTS is 17–39%[8,9] and the heritability of CTS has been estimated to be 0.46 in women[10]. The largest reported genome-wide association study (GWAS) of CTS yielded 16 risk loci using data from 12,312 cases and 389,344 controls from the UK[11]. Here, we expand the previous study by combining 48,843 cases and 1,190,837 controls from Iceland (deCODE genetics), the UK (UK Biobank), Denmark (Danish Blood Donor Study [DBDS] and Copenhagen Hospital Biobank [CHB]), and Finland (Finngen) in a GWAS meta-analysis of CTS, and find 53 independent sequence variants at 50 loci, of which 32 variants have not been previously reported.

## Results

**GWAS meta-analysis.** In a meta-analysis, we combined CTS GWAS results for 37.1 million sequence variants using 48,843 cases and 1,190,837 controls from Iceland, the UK, Denmark, and Finland (Table 1, Supplementary Table 1). We applied logistic regression to test for the association of sequence variants with CTS under an additive model, and combined the GWAS summary results of each dataset in a fixed-effects inverse variance model. To account for multiple testing, we used weighted genome-wide significance thresholds based on the predicted functional impacts of the associated variants[12] (Supplementary Table 2). Sequence variants at 50 loci associated with CTS (Fig. 1 and Supplementary Data 1). Conditional analyses revealed secondary signals at three of the loci (Supplementary Data 2 and Supplementary Fig. 1).

Of the 53 variants, 32 represent novel associations with CTS while 21 have previously been associated with CTS[11,13], directly or through a correlated variant ($r^2 \geq 0.2$) at the corresponding locus in our results (Supplementary Data 3). At six of the 21 reported loci, we found associating coding variants, an in-frame deletion, three splice region variants, and two missense variants, while all previously reported variants at these loci are non-coding (Supplementary Data 3). To test the 21 previously reported variants in our meta-analysis, we re-ran meta-analyses excluding the datasets that overlap with the discovery datasets. One of the variants was only present in the Finnish dataset (EAF = 0.10%) and thus could not be validated. Of the 20 remaining variants, we found support for 19 signals with the same direction of effect in the datasets tested ($P \leq 0.05$; Supplementary Data 3). The effect estimates of the 53 CTS variants identified in the meta-analysis were in the same direction in all four datasets (Supplementary Fig. 3) and there was no evidence of heterogeneity (all $P$-het $\geq 0.05/53 = 9.4 \times 10^{-4}$). The UK dataset was the largest and 10 of the 53 variants reached the genome-wide significance in that dataset. Of the 53 variants, 5 are low-frequency variants (EAF < 5%), including the missense variant (p.Glu366Lys) in *SERPINA1*, which has the largest effect on CTS ($P = 2.9 \times 10^{-24}$, OR = 0.76). We constructed a polygenic risk score (PRS) based on effect estimates from the meta-analysis, excluding the UK dataset, and predicted into the UK dataset. The PRS explained 2% of the variance. SNP heritability estimated using linkage disequilibrium (LD) score regression[14], was 0.19 (95% CI 0.11–0.27) in Iceland, 0.34 (95% CI 0.30–0.38) in the UK, and 0.16 (95% CI 0.11–0.20) in Denmark.

**Functional annotation.** To search for causal genes associated with CTS at each locus (*i.e.* mediating the effect of the variant on CTS), we annotated the 53 CTS sequence variants and correlated variants ($r^2 \geq 0.8$ and within ± 1MB). Of these, 35 were predicted to affect coding or splicing of a protein (N = 17, Supplementary Data 4), mRNA expression (top cis expression quantitative trait loci [*cis*-eQTL]; N = 24, Supplementary Data 5), and/or plasma protein levels[15] (top protein quantitative trait loci [*cis* and *trans*-pQTL]; N = 10, Supplementary Data 6). Of the 35 CTS variants and correlated variants, 22 had evidence that pointed to a single gene (Fig. 2a and Supplementary Data 4–6). To estimate the deleteriousness of the 53 variants, we used a scaled Combined Annotation Dependent Depletion (CADD) score[16], which ranks variants based on predicted pathogenicity, and found that 19 variants are among the top 10% of pathogenic variants, and 10 variants were in the top 1% (Fig. 2a and Supplementary Data 7). In addition, we performed a gene-set enrichment analysis in FUMA[17] based on the genes at the 50 loci and found a strong enrichment for extracellular matrix components ($P_{adj} = 4.1 \times 10^{-11}$; Fig. 2b and Supplementary Data 8).

**Table 1 Summary of the GWAS datasets in the CTS meta-analysis.**

| Population | GWAS dataset | Sequence variants | Cases | Controls |
|---|---|---|---|---|
| Iceland | deCODE genetics | 20,697,529 | 8,122 | 318,161 |
| The UK | UK Biobank | 32,381,502 | 19,849 | 411,179 |
| Denmark | Copenhagen Hospital Biobank and The Danish Blood Donor Study | 27,776,695 | 9,664 | 266,450 |
| Finland | Finngen | 12,830,475 | 11,208 | 195,047 |
| | Total | | 48,843 | 1,190,837 |

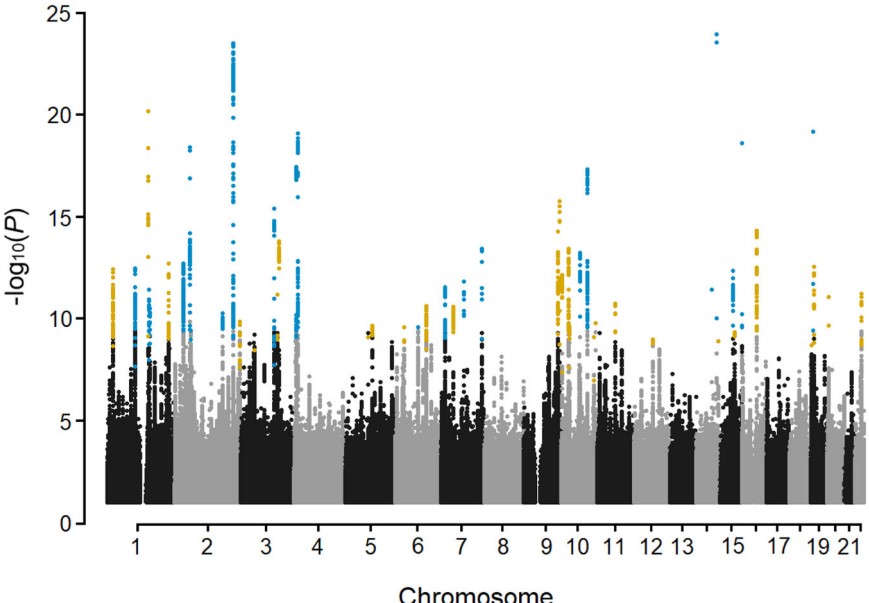

**Fig. 1 A Manhattan plot of the CTS meta-analysis results, highlighting 50 loci.** The -log$_{10}$P-values (y-axis) are plotted for each variant against their chromosomal position (x-axis). Variants with P-values below their weighted variant-class threshold are highlighted. Variants that have not been previously reported are marked in orange and previously reported variants are marked in blue. P-values are two-sided and derived from a likelihood-ratio test. Manhattan plots for the different datasets used are shown in Supplementary Fig. 2.

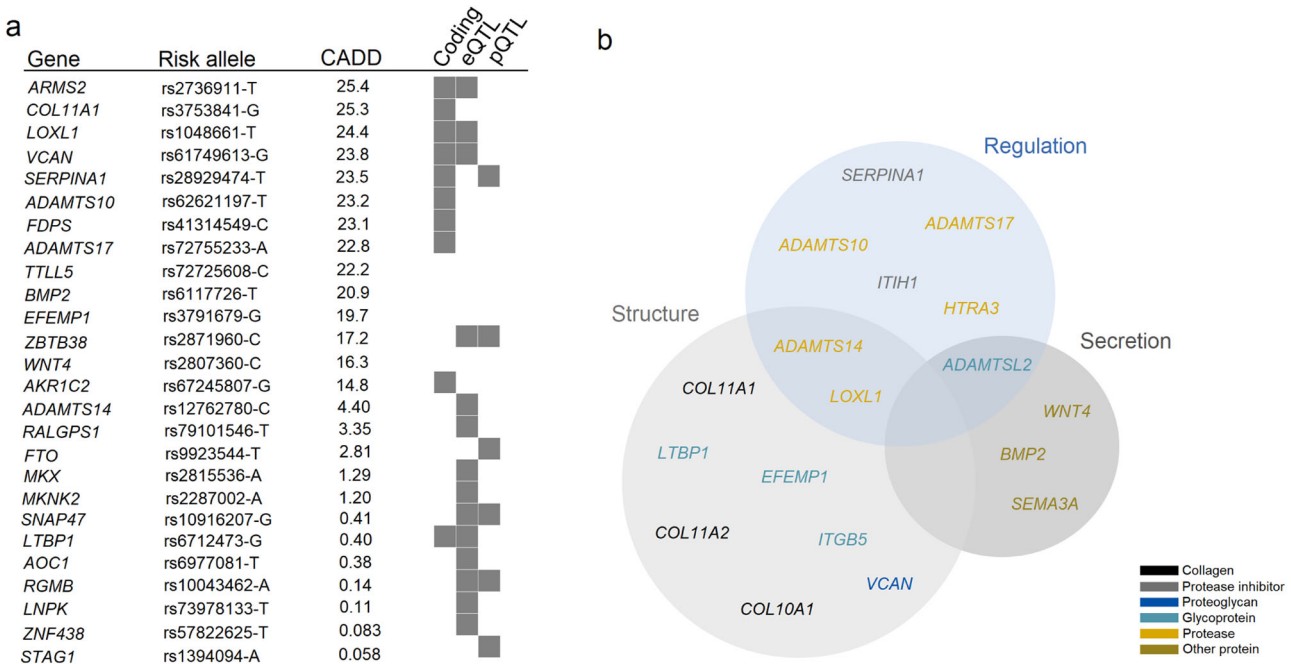

**Fig. 2 Functional annotation and gene-set analysis. a** Variants (N = 22) with evidence pointing to a single effect-mediating gene in the form of protein-coding or splicing variants, variants affecting mRNA expression (top cis-eQTL) or plasma protein levels (pQTL), and variants (N = 4) with evidence of pathogenicity (CADD score ≥ 12.4, suggested threshold for pathogenicity[64]). **b** Gene-set enrichment analysis revealed an enrichment for genes encoding proteins that are components of the extracellular matrix. The Venn diagram divides the implicated genes into groups based on the functional role of the encoded protein in the extracellular matrix; regulation (blue), secretion (dark gray), and/or structure (light gray). Genes encoding proteins in different protein categories are colored black (collagens), gray (protease inhibitors), navy (proteoglycans), turquoise (glycoproteins), orange (proteases), and bronze (other proteins).

**CTS variant effect in ulnar and radial nerve lesions.** Although not passing through the carpal tunnel, the ulnar and radial nerves course through the forearm and wrist to help co-ordinate the movement of the forearms and hands. Using data from Iceland, the UK, Denmark, and Finland, we performed meta-analyses of lesions of the ulnar nerve (9429 cases and 1,136,861 controls) and the radial nerve (1645 cases and 1,232,006 controls) to study the effect of the 53 CTS variants in these phenotypes. None of the

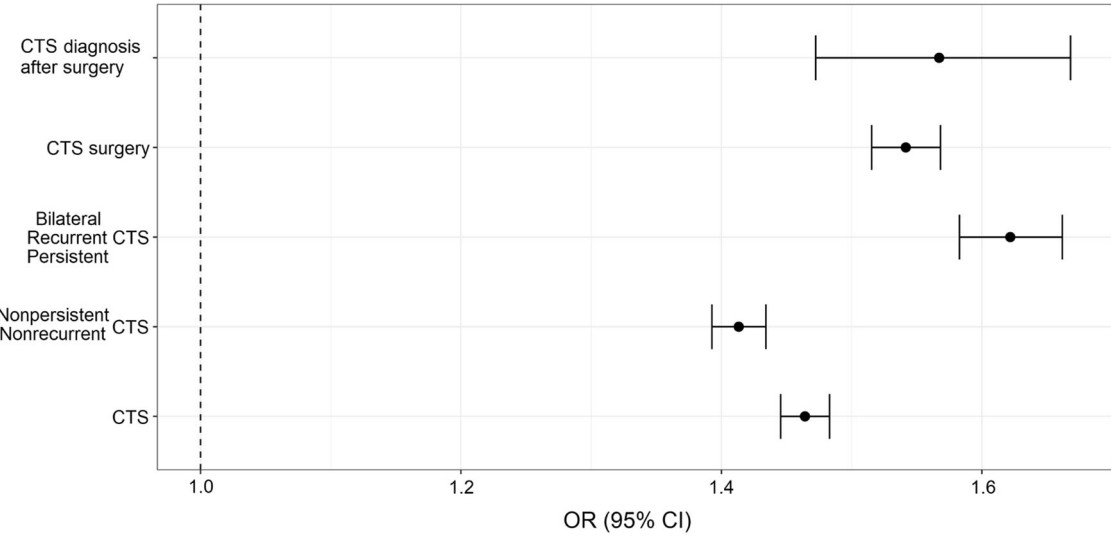

**Fig. 3 Genetic propensity for CTS in suggested severity groups.** The results from the PRS analyses were meta-analyzed[58] (see Supplementary Fig. 4 for each dataset). The point estimate of the OR and 95% CI (error bars) adjusted for genomic inflation on x-axis, were estimated in each group, y-axis.

variants associated with lesions of the ulnar or the radial nerve (Supplementary Data 9 and 10).

**Polygenic risk score and sex effect.** The most common causes of recurrent or persistent CTS include incomplete surgical release of the transverse carpal ligament, fibrous proliferation, and recurrent tenosynovitis. We constructed a CTS PRS using LDpred[18] to estimate whether the level of genetic propensity varies in differently defined groups of CTS cases based on suggested clinical severity. We constructed the PRS for individual datasets (except the Finnish dataset for which we did not have individual level data) using summary data from the meta-analysis, leaving each dataset out in turn. Using the CTS PRS, we estimated the effect size in five groups defined as (1) all CTS cases ($N = 37,635$), (2) nonrecurrent or nonpersistent CTS cases ($N = 28,200$), (3) bilateral, recurrent, or persistent CTS cases ($N = 9,435$), (4) CTS cases that have undergone surgery ($N = 18,281$), and (5) CTS cases that were diagnosed with CTS after surgery ($N = 2,843$). The bilateral, recurrent, or persistent cases (Iceland = 21.6%, the UK = 26.2%, and Denmark = 27.7%; Supplementary Table 1) were defined as cases with six or more months between healthcare encounters resulting in a CTS diagnosis. As expected, bilateral, recurrent, or persistent CTS cases had higher PRS than nonrecurrent or nonpersistent cases ($P$-het = $1.5 \times 10^{-22}$; Fig. 3 and Supplementary Fig. 4). We did not observe a significant difference between CTS surgery and CTS diagnosis after surgery.

CTS is more commonly reported among females than males and studies have shown that the carpal tunnel cross-sectional area relative to the size of the hand is smaller in females[19]. In the CTS study sample, 34% were males and 66% females (Supplementary Table 1) and the average age of diagnosis was 57 years for males and 55 years for females. Further, the recurrence was 25.4% in males and 27.1% in females. In light of reported sex differences in risk of CTS, we applied sex-specific models for the Icelandic, the UK, and Danish datasets for the 53 CTS variants (Supplementary Data 11), but no variant had an effect that significantly differed between the sexes after accounting for multiple testing ($P \geq 0.05/53 = 9.4 \times 10^{-4}$).

**Low-frequency variant in *SERPINA1*.** The variant most significantly associated with CTS is a low-frequency missense variant (p.Glu366Lys) in *SERPINA1*, present in all four datasets (rs28929474-T, EAF = 1.86%, OR = 0.76, $P = 2.9 \times 10^{-24}$).

*SERPINA1* encodes α-1 antitrypsin and rs28929474-T has been reported to cause α-1 antitrypsin deficiency[20]. It has also been widely reported in the literature to associate with a broad range of traits[21–26], including multiple plasma proteins. Our results show that the variant affects 36 plasma proteins (Supplementary Data 6). The only variant in high LD ($r^2 \geq 0.8$) with rs28929474 is a downstream variant, rs112635299-T (Fig. 4).

**Genetic correlation and Mendelian randomization.** We performed genetic correlation analyses using the CTS meta-analysis and summary statistics from 1319 GWASs ($P \leq 0.05/1319 = 3.8 \times 10^{-5}$, Supplementary Data 12)[27]. We excluded the UK dataset from the meta-analysis as it overlaps with the data used in most of the summary statistics. Substantial genetic correlations have previously been reported between CTS and anthropometric traits including height and BMI[11]. Our collective data of genetic correlation (Supplementary Data 12) and Mendelian randomization (MR; Supplementary Fig. 5 and Supplementary Table 3), using 3290 variants previously associated with height[28] and 941 variants previously associated with BMI as instrumental variables, further support these findings. However, in contrast to the previous study, our results show evidence of confounding by unbalanced genetic pleiotropy for both height (intercept = −0.003, $P = 1.1 \times 10^{-11}$) and BMI (intercept = 0.006, $P = 2.2 \times 10^{-16}$). Of the genes reported in the CTS meta-analysis, 22 have previously been associated with height and 12 with body fat distribution (Supplementary Data 8). BMI is commonly used as a proxy measurement of body adiposity. However, BMI does not discriminate between adipose and lean mass, and between fat stored in different compartments of the body. Impedance measurements address this problem to some extent. CTS correlates genetically with both fat percentage and fat-free mass of arms, legs, and trunk based on LD score regression (Supplementary Data 12). Besides anthropometric traits, the strongest genetic correlation is between CTS and age when hormone-replacement therapy was started ($r_g = -0.48$, $P = 1.4 \times 10^{-12}$), other and unspecified osteoarthritis defined by ICD-10 code M19, which includes primary osteoarthritis of the wrists ($r_g = 0.68$, $P = 5.7 \times 10^{-11}$), and feeling restless ($r_g = 0.55$, $P = 5.4 \times 10^{-10}$).

## Discussion
Here, we report a GWAS meta-analysis of CTS that combines 48,843 cases from four populations and expands previous studies

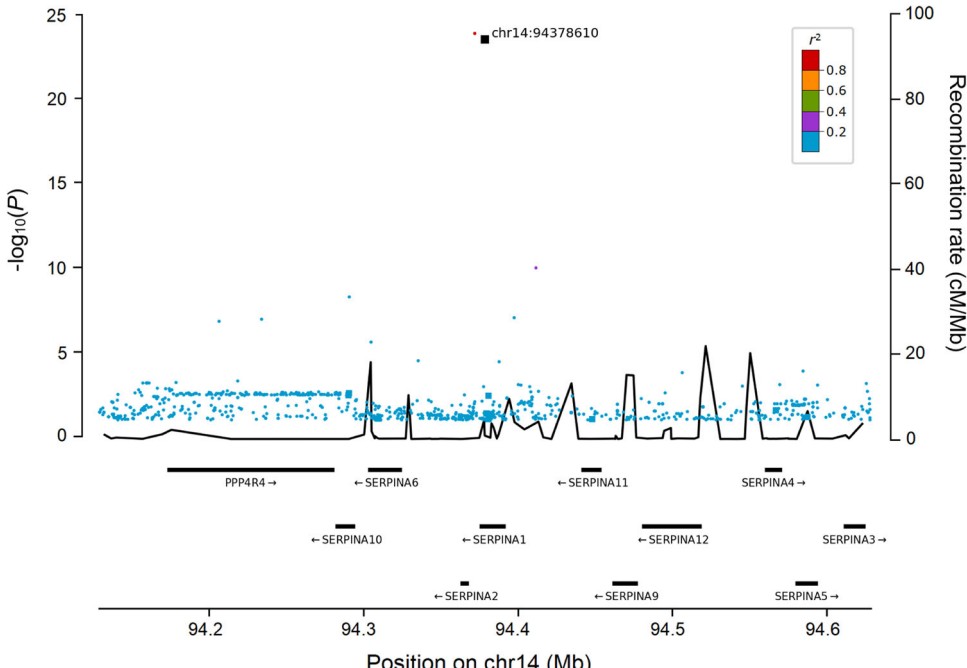

**Fig. 4 Regional plot of the *SERPINA1* locus.** Variants are colored by the degree of correlation ($r^2$) with the lead variant, which is colored black. Moderate impact variants are represented by squares. The $-\log_{10}P$-values on the left y-axis (two-sided logistic regression) are plotted for each variant against their chromosomal position (x-axis). The right y-axis shows calculated recombination rates based on the Icelandic data at the chromosomal location, plotted as solid black lines.

by identifying 53 variants at 50 loci. Among these 53 variants is a low-frequency variant in *SERPINA1* that protects against CTS. Through fine-mapping, eQTL, pQTL, CADD score, and gene-set enrichment analysis, we highlighted potential effect-mediating genes and the biological roles of a large portion of the variants. Using PRS, we estimated the genetic propensity in differently defined severity groups of CTS and showed that bilateral, recurrent, or persistent CTS cases had higher PRS than nonrecurrent or nonpersistent cases. There was a strong genetic correlation between CTS and early hormonal-replacement therapy, osteoarthritis, and feeling restless. Further, genetic correlation studies of anthropometric traits were consistent with previous findings but MR analyses showed evidence of genetic pleiotropy when using height and BMI variants as instrumental variables.

Based on functional annotation, we highlighted 35 genes that may participate in the pathogenesis of CTS. Three of the candidate causal genes are *COL11A1*, *COL11A2*, and *COL10A1*. These genes are members of the collagen family (COL) and encode collagens, the most abundant proteins in the extracellular matrix where they give structural support to resident cells and account for 90% of bone matrix protein content[29]. *COL11A1* has previously been implicated in CTS[30] and *COL11A1* and *COL11A2* have been implicated in tendinopathies[31–33]. Mutations and abnormal expression of *COL10A1* have been linked to abnormal chondrocyte hypertrophy and osteoarthritis in humans[34–36], and *Col10a1* knockout mice have a reduced thickness of growth plate and articular cartilage, altered bone content, and atypical distribution of matrix components within growth plate cartilage[37]. In our study, a variant in high LD ($r^2 = 0.98$) with the CTS variant in *COL10A1*, associates with Collagen α-1(X) chain in plasma ($P = 4.0 \times 10^{-180}$, $\beta = -0.25$; Supplementary Data 6), suggesting a decrease of the protein in CTS cases.

Collagens are substrates for disintegrin and metalloproteases with thrombospondin motifs (ADAMTS)[38] and members of the lysyl oxidase (LOX) family[39]. Three of the candidate causal genes highlighted in the study encode ADAMTS (*ADAMTS10*,

*ADAMTS14*, and *ADAMTS17*) and one encodes ADAMTS-like protein (*ADAMTSL2*), which resembles ADAMTS in structure but lacks proteolytic activity. ADAMTS have regulatory roles and are involved in the processing of procollagens to mature collagens and control the structure and function of the extracellular matrix[40]. Another collagen substrate and a highlighted gene, *LOXL1*, provides collagen fibers their tensile strength and structural integrity by oxidizing lysine residues, resulting in the covalent cross-linking and stabilization of these structural components[39]. The missense variant (p.Arg141Leu) in *LOXL1* is among the top 1% of pathogenic variants (CADD score = 24.4). As ADAMTS process the COL substrates which are stabilized by LOX, we assume that these processes are inter-linked and speculate that distortion of one may affect the others and thus, increase the risk of CTS.

Other genes encoding components of the extracellular matrix that were highlighted in the study include *SERPINA1* and *ITGB5*. The missense variant (p.Glu366Lys) in *SERPINA1* is the top signal in the meta-analysis and has a strong protective effect on CTS and also associates with the plasma levels of multiple proteins (Supplementary Data 6) supporting previously reported observations. *ITGB5* encodes a subunit of integrin which participates in cell adhesion in the extracellular matrix, and is upregulated in CTS cases[41]. The in-frame deletion reported here associates with plasma levels of several proteins (Supplementary Data 6) that are involved in collagen processes and immune response among other functions.

Using CTS PRS, we found differences between nonrecurrent or nonpersistent and bilateral, recurrent, or persistent CTS cases, indicating higher genetic propensity for CTS among the latter group. We did not have data on which hand was affected and could not define bilateral CTS cases specifically. Thus, we combined cases that had two or more carpal tunnel flare ups with six or more months between healthcare encounters and compared them to cases with one carpal tunnel flare up.

Entrapment neuropathies of the median, ulnar, and radial nerves are characterized by alteration of nerve function that are

caused by mechanical or dynamic compression[1]. None of the variants associating with CTS showed strong evidence of association with lesions of the ulnar or radial nerve, suggesting a specific pathology in CTS cases.

CTS is common and debilitating. Our study identified genome-wide associations with CTS and revealed sequence variants that point to genes that may contribute to the pathogenesis. Our results suggest that the components of the extracellular matrix play a key role in the pathogenesis of CTS by putting pressure on the median nerve at the carpal tunnel.

## Methods

**Study sample and ethics statement**. A large fraction of the Icelandic population has participated in a nationwide research program at deCODE genetics. Participants donated blood or buccal samples after signing a broad informed consent allowing the use of their samples and data in all projects at deCODE genetics approved by the National Bioethics Committee (NBC). The data in this study was approved by the NBC (VSN-19-158; VSNb2019090003/03.01) following review by the Icelandic Data Protection Authority. All personal identifiers of the participants' data were encrypted in accordance with the regulations of the Icelandic Data Protection Authority. The Icelandic CTS cases were obtained in collaboration with Icelandic physicians at Landspitali—National University Hospital in Reykjavik, the Registry of Primary Health Care Contacts, and the Registry of Contacts with Medical Specialists in Private Practice. The CTS cases were identified using International Classification of Diseases 10 (ICD-10) code G56.0, ICD-9 code 354.0, and Nomseco Classification of Surgical Procedures (NCSP) code ACC51 (decompression and freeing of nervus medianus) through the scrutiny of hospital records from 1985 to 2020. Cases with lesions of the ulnar and radial nerves were identified using ICD-10 code G56.2 and G56.3, respectively.

The UK Biobank resource includes extensive phenotype and genotype data from ~500,000 participants in the age range of 40 to 69 from across the UK that have provided an informed consent[42]. The North West Research Ethics Committee reviewed and approved UK Biobank's scientific protocol and operational procedures (REC Reference Number: 06/MRE08/65). This study was conducted using the UK Biobank resource under application number 24898. CTS cases were identified by searching for ICD-10 code G56.0 and Classification of Interventions and Procedures (OPCS) codes A651 (carpal tunnel release) and A692 (revision of carpal tunnel release) in General Practice clinical event records (Field ID 42040) and UK hospital diagnoses (Field ID 41270 and 41271).

The Copenhagen Hospital Biobank (CHB) is a research biobank, which contains left over samples from diagnostic procedures on hospitalized and outpatients in the Danish Capital Region hospitals. Under the "Genetics of pain and degenerative diseases" protocol, approved by the Danish Data Protection Agency (P-2019-51) and the National Committee on Health Research Ethics (NVK-18038012), we identified CTS cases by ICD-10 code G56.0 and NCSP code ACC51. The Danish Blood Donor Study (DBDS) Genomic Cohort is a nationwide study of ~110,000 blood donors[43]. The Danish Data Protection Agency (P-2019-99) and the National Committee on Health Research Ethics (NVK-1700407) approved the studies under which genetic data on DBDS participants were obtained. The DBDS data requested for this study was approved by the DBDS steering committee.

The FinnGen database consists of samples collected from the Finnish biobanks and phenotype data collected at the national health registers. The Coordinating Ethics Committee of the Helsinki and Uusimaa Hospital District evaluated and approved the FinnGen research project. The project complies with existing legislation (in particular the Biobank Law and the Personal Data Act). The official data controller of the study is the University of Helsinki. Subjects were identified using ICD-10 code G56.0 and ICD-9 code 354.0. NCSP-FI codes (Finnish NCSP adaption) were not available. The summary statistics for CTS were imported on May 11th, 2021 from a source available to consortium partners (version 5; http://r5.finngen.fi).

Genetic ancestry quality control was performed for the Icelandic[44], British[45], and Danish[46] participants. All participants were genotypically verified as being of European descent.

In total, we had 48,843 carpal tunnel cases (8,122 from Iceland, 19,849 from the UK, 9,664 from Denmark, and 11,208 from Finland) and 1,190,837 controls (318,161 from Iceland, 411,179 from the UK, 266,450 from Denmark, and 195,047 from Finland) in the meta-analysis.

**Genotyping and imputation**. The preparation of the Icelandic samples, genotyping, whole-genome sequencing (WGS), and imputation were performed at deCODE genetics[44,47]. The genomes of 49,962 Icelanders were WGS using GAIIx, HiSeq, HiSeqX, and NovaSeq Illumina technology to a mean depth of at least 17.8×. Single nucleotide polymorphisms (SNPs) and insertions/deletions (indels) were identified and their genotypes were called using joint calling with Graphtyper[48]. In addition, over 166,000 Icelanders (including all sequenced Icelanders) have been genotyped using various Illumina SNP chips and their genotypes phased using long-range phasing[49], which allows for improving genotype

calls using haplotype sharing information. Subsequently, genealogic information was used to impute sequence variants into the chip-typed Icelanders, as well as their first- and second-degree relatives[50] to increase the sample size and power for association analysis.

The UK Biobank samples were genotyped with a custom-made Affymetrix chip, UK BiLEVE Axiom in the first 50,000 individuals[51], and the Affymetrix UK Biobank Axiom array[52] in the remaining participants. Samples were filtered on 98% variant yield and any duplicates removed. Over 32 million high-quality sequence variants and indels to a mean depth of at least 20× were identified using Graphtyper[48]. Quality-controlled chip genotype data were phased using Shapeit4[53]. Variants where at least 50% of the samples had a GQ score > 0 were used to prepare a haplotype reference panel using in-house tools and the long-range phased chip data. The haplotype reference panel variants were then imputed into the chip genotyped samples using in-house tools and methods described above for the Icelandic data[44,49].

Samples from 276,114 Danes from the CHB and DBDS were genotyped using Illumina Global Screening Array chips and long-range phased together with ~238,000 genotyped samples from North-western Europe using Eagle[54]. Samples and variants with less than 98% yield were excluded. A haplotype reference panel was prepared in the same manner as for the Icelandic and UK data[44,49] by phasing whole-genome sequence genotypes of 15,576 individuals from Scandinavia, the Netherlands, and Ireland using the phased chip data. Graphtyper was used to call the genotypes which were subsequently imputed into the phased chip data. Whole genome sequencing, chip-typing, quality control, long-range phasing, and imputation from which the data for this analysis were generated was performed at deCODE genetics.

A custom-made FinnGen ThermoFisher Axiom array (>650,000 SNPs) was used to genotype ~177,000 FinnGen samples at Thermo Fisher genotyping service facility in San Diego. Genotype calls were made with AxiomGT1 algorithm. Individuals with ambiguous gender, high genotype missingness (>5%), excess heterozygosity (±4 SD), and non-Finnish ancestry were excluded. Variants with high missingness (>2%), low Hardy-Weinberg equilibrium ($< 1 \times 10^{-6}$), and minor allele count (<3) were excluded. High coverage (25–30×) WGS data were used to develop the Finnish population-specific SISu v3 imputation reference panel with Beagle 4.1. More than 16 million variants have been imputed (https://finngen.gitbook.io/documentation/methods/genotype-imputation).

**Association analysis**. We applied logistic regression using the Icelandic, UK, and Danish data and combined the results with imported association results from Finland (https://r5.finngen.fi/) to test for association between sequence variants and CTS. For the additive model, the expected allele counts were used as a covariate. We used LD score regression to account for distribution inflation due to cryptic relatedness and population stratification[14] and used the intercepts as correction factors (CF) in the Icelandic (CF = 1.19), UK (CF = 1.03), and Danish (CF = 1.03) datasets.

In the Icelandic association analysis, we adjusted for sex, county of origin, age at data analysis or age at death (first and second order terms included), blood sample availability for the individual, and an indicator function for the overlap of the lifetime of the individual with the time span of phenotype collection. In the UK association analysis, we adjusted for sex, age, and the first 20 principal components[45]. In the Danish association analysis, we adjusted for sex, whether the individual had been chip-typed and/or sequenced, and the first 20 principal components. The imported Finnish association analysis was adjusted for sex, age, the genotyping batch, and the first 10 principal components.

We combined CTS GWASs from Iceland, the UK, and Denmark with summary statistics from Finland using a fixed-effects inverse variance method[55] based on effect estimates and standard errors in which each dataset was assumed to have a common OR but allowed to have different population frequencies for alleles and genotypes. The total number of variants included in the meta-analysis that had imputation information above 0.8 and MAF > 0.01% was 37,071,338 (20,697,529 in Iceland, 32,381,502 in the UK, 27,776,695 in Denmark, and 12,830,475 in Finland). We estimated the genome-wide significance threshold using a weighted Bonferroni adjustment that controls for the family-wise error rate[12] (Supplementary Table 2). Sequence variants were mapped to NCBI Build38 and matched on position and alleles to harmonize the four datasets. Variants were weighted based on predicted functional impact[12] (Supplementary Table 2). Manhattan plots were generated using qqman package in R[56].

Conditional association analyses were performed on the GWASs from Iceland, the UK, and Denmark using true imputed genotypes of participants. Approximate conditional analyses (COJO), implemented in the GCTA-software[53], were applied on the lead variants in the Finnish summary statistics. The analyses were restricted to variants within 1 Mb from the index variants. LD between variants was estimated using a set of 5,000 WGS Icelanders. After adjusting for all variants in high LD ($r^2 > 0.8$) and vice versa, the P-values were combined for all four datasets to identify the most likely causal variant at each locus and any secondary signals. Based on the number of variants tested, we chose a conservative P-value threshold of $<5 \times 10^{-8}$ for secondary signals.

**Polygenic risk score and heritability**. Polygenic risk score (PRS) was generated using a set of 611,000 high quality variants across the genome were used to avoid

uncertainty due to imputation quality[57]. LD estimated from almost 15,000 phased Icelandic samples, was used derive adjusted effect estimates by applying LDpred[18]. The effect estimates were used as weights. We generated PRS into the Icelandic, the UK, and Danish datasets and used leave-one-out meta-analyses, where the summary data from that particular dataset was omitted, to avoid any bias in PRS estimates. In addition, we meta-analyzed the results from the dataset-specific PRS analyses using a random-effects model, fitted via maximum likelihood estimation[58]. The unit of the effect for each PRS is in SD. Since the effects on CTS were comparable (overlapping confidence intervals), we performed no further scaling.

We estimated the SNP heritability for the combined CTS GWASs from Iceland, the UK, and Denmark using LD score regression[14] and precomputed LD scores based on about 1.1 million variants from European ancestry samples (downloaded from: https://data.broadinstitute.org/alkesgroup/LDSCORE/eur_w_ld_chr.tar.bz2).

**Genetic correlation**. Genetic correlation analyses between the CTS meta-analysis and 1,319 published GWAS traits ($P \leq 3.8 \times 10^{-5}$) from the UK Biobank[27] with effective sample size over 5,000 were performed using LD score regression[14,59], which suggests the minimal effective sample size of 5,000 for each trait to get unbiased estimates of genetic correlation and heritability. Since participants in the published GWAS studies are of Caucasian ancestry, we used pre-computed LD scores from a 1000 genome panel with $r^2$ from HapMap3, excluding HLA region. The HLA region was excluded for its genetic complexity and association with a wide number of traits. The default parameters of the LD score regression were used to compute the genetic correlation and heritability estimates.

**Mendelian randomization**. A two sample MR analysis was performed to estimate the causal relationship between CTS and exposure traits, that we used as instrumental variables. We used the inverse-variance-weighted (IVW) method to estimate the causal relationship between variables, a t-test to compute the $P$-value, and the Egger method to test for pleiotropy in IVW estimates, all implemented in the MendelianRandomization package[60] in R.

**Functional data**. To highlight potential causal genes associating with CTS, we annotated the CTS associations or variants in high LD ($r^2 \geq 0.8$ and within $\pm$ 1MB) that are predicted to affect coding or splicing of a protein (variant effect predictor using Refseq gene set), mRNA expression (top local expression quantitative trait loci [cis-eQTL]) in multiple tissues from deCODE, GTEx (https://www.gtexportal.org/home/), and other public datasets, and/or plasma protein levels (top protein quantitative trait loci [pQTL]).

RNA sequencing was performed on whole blood from 13,175 Icelanders and on subcutaneous adipose tissue from 750 Icelanders, described in detail elsewhere[61]. Gene expression was computed based on personalized transcript abundances using kallisto[62]. Association between sequence variants and gene expression (cis-eQTL) was estimated using a generalized linear regression, assuming additive genetic effect and quantile normalized gene expression estimates, adjusting for measurements of sequencing artefacts, demographic variables, blood composition, and hidden covariates[63].

We used the SomaLogic® SOMAscan proteomics assay to test the association of the sequence variants with protein levels in plasma[15]. The assay scanned 4,907 aptamers that measure 4,719 proteins in samples from 35,559 Icelanders with genetic information available at deCODE genetics. Plasma protein levels were standardized and adjusted for year of birth, sex, and year of sample collection (2000–2019).

We performed gene-based enrichment analysis using the GENE2FUNC tool in FUMA[17]. The genes were tested for over-representation in different gene sets, including Gene Ontology cellular components (MsigDB c5) and GWAS catalogue-reported genes.

**Reporting summary**. Further information on research design is available in the Nature Research Reporting Summary linked to this article.

## Data availability

The GWAS summary statistics for the CTS meta-analysis are available at https://www.decode.com/summarydata/. Other data generated or analyzed in this study are included in the article and its Supplementary data and information.

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

## Acknowledgements

We thank the participants in this study for their valuable contribution to the research. We thank all investigators and colleagues who contributed to data collection, phenotypic characterization of clinical samples, genotyping, and analysis of the whole-genome association data. This research was conducted using the UK Biobank Resource (application number 24898). We acknowledge the participants and investigators of the Finn-Gen study. The financial support from the European Commission to the painFACT project, TET, (H2020-2020-848099) is acknowledged.

## Author contributions

A.T.S., G.B., H.S., and K.S. designed the study. A.T.S., G.B., E.F., G.E., L.S., M.S.N., S.H.M., D.F.G., P.S., T.E.T., H.S., and K.S. analyzed the data and interpreted the results. Data collection and subject ascertainment and recruitment were carried out by G.B., G.B.W., A.O., A.B., O.A.S., A.V., T.F.H., R.L.J., C.E., M.S., S.B., K.B., S.R.O., A.T., C.H., O.B.P., D.B.D.S., G.C., and I.J. A.T.S. drafted the manuscript with input and comments from G.B., E.F., G.E., L.S., M.S.N., T.A.O., S.S., G.B.W., A.B., D.F.G., P.S., T.E.T., H.S., and K.S. All authors contributed to the final version of the manuscript.

## Competing interests

A.T.S., G.B., E.F., G.E., L.S., M.S.N., T.A.O., S.S., G.B.W., S.H.M., A.O., I.J., D.F.G., P.S., T.E.T., H.S., and K.S. are employees of deCODE genetics/Amgen Inc. The remaining authors declare no competing interests.

## Additional information

## DBDS Genetic Consortium

Steffen Andersen[14], Karina Banasik[6], Søren Brunak[6], Kristoffer Burgdorf[7], Maria Didriksen[7], Khoa Manh Dinh[8], Christian Erikstrup[8], Daniel F. Gudbjartsson[1,13], Thomas Folkmann Hansen[5,6], Henrik Hjalgrim[15], Gregor Jemec[16], Poul Jennum[17], Pär Ingemar Johansson[7], Margit Anita Hørup Larsen[7],

Susan Mikkelsen[8], Kasper Rene Nielsen[18], Mette Nyegaard[19], Sisse Rye Ostrowski [7,9], Ole Birger Pedersen [12✉], Hreinn Stefánsson[1], Susanne Sækmose[12], Erik Sørensen[7], Unnur Thorsteinsdottir[1,2], Mie Topholm Bruun[20], Henrik Ullum[21], Thomas Werge[22] & Kari Stefansson [1,2✉]

[14]Department of Finance, Copenhagen Business School, Copenhagen, Denmark. [15]Department of Epidemiology Research, Statens Serum Institut, Copenhagen, Denmark. [16]Department of Clinical Medicine, Zealand University Hospital, Roskilde, Denmark. [17]Department of Clinical Neurophysiology, , University of Copenhagen, Copenhagen, Denmark. [18]Department of Clinical Immunology, Aalborg University Hospital, Aalborg, Denmark. [19]Department of Biomedicine, Aarhus University, Aarhus, Denmark. [20]Department of Clinical Immunology, Odense University Hospital, Odense, Denmark. [21]Statens Serum Institute, Copenhagen, Denmark. [22]Institute of Biological Psychiatry Mental Health Centre Sct. Hans Copenhagen University Hospital, Roskilde, Denmark.

