## [Peer Review File · Nature Communications]

A genome-wide meta-analysis identifies 50 genetic loci associated with carpal tunnel syndromeReviewers' Comments:

Reviewer #1:

Remarks to the Author:

Nature Communications peer review report

A genome-wide meta-analysis identifies 53 sequence variants associating with carpal tunnel syndrome

Skuladottir et al.

This article describes a GWAS meta-analysis of Carpal Tunnel Syndrome. The authors discovered 53 variants at 50 loci, 32 of which have not previously been.

I enjoyed reading the paper. The work is of high quality, and will be of interest to the scientific community. The paper overall is well-written. I have suggested changes to improve clarity of the manuscript, and I suggest acceptance after these comments are addressed.

Major comment

I have not seen a data availability statement. It is vital that full summary statistics (not just the significant results presented in supplementary data) from these analyses are deposited in an appropriate public repository so they can be further used by the scientific community. This should be ensured by the editors prior to acceptance for publication.

Minor Comments

Title

1. Title – replace associating with associated

Introduction

2. "contains the carpal bones, the transverse carpal ligament," – the carpal tunnel is bounded by these structures (they make up the anatomical boundary) it does not contain them.
3. "The median nerve innervates muscles of the hand" change to "The median nerve innervates the thenar muscles and radial lumbricals" as the ulnar nerve innervates most of the intrinsic muscles of the hand.
4. Diagnosis is based on a typical medical history in the first instance.
5. remove "i.e. nerve conduction velocity across the carpal tunnel" – electrophysiology is much more than conduction velocity.
6. Treatment usually consists of lifestyle changes, night-time splintage, steroid injection, and surgical decompression.
7. "sequence variants at 50 loci, thereof 32 novel." Change to "sequence variants at 50 loci, of which 32 have not been previously reported".

Methods

The methods are appropriate and comprehensive.

8. We estimated the genome-wide significance threshold using a weighted Bonferroni adjustment that controls for the family-wise error rate¹². Please refer to Supplementary table 2 here

Results

9. Figure 1 should be much larger on full submission, as it is very hard to interpret. Please highlight the previously unreported and the previously reported associations in a different colour from each other.
10. Table 2 – please highlight in bold loci with previously undescribed associations
11. Table 2 – from which population is the reported EAF%? Please note in table legend
12. “the top 10% of pathogenic variants, thereof 10 in the top 1%” – change to “the top 10% of pathogenic variants, and 10 variants were in the top 1%”
13. PRS analysis: the definition of recurrent CTS is incorrect – this is a further operation not a recurrence. Could this be that people have had surgery on their left carpal tunnel and then >6months later had surgery on their right carpal tunnel? If you do not have data on side of surgery, recurrent should be limited to those with three or more surgical episodes (as no-one has three arms!), and the two or more surgical episodes (your group 3) should be redefined as either bilateral (most of these cases I think) or recurrent. The true recurrence rate in England is around 3% at 10 years (see figure 3 in doi: 10.1016/S2665-9913(20)30238-1). Please change this and Supplementary table 1.

In the conclusion you state: “We did not have data on which hand was affected and could not define bilateral CTS cases specifically. Thus, we were not able to exclude the possibility of CTS in each hand at different timepoints in a case defined as recurrent.” This means it is indeed essential you change the nomenclature here. The vast majority of these cases will be bilateral, NOT recurrent.

14. PRS: how did you define “symptoms after surgery”? The cause of such symptoms can be very broad (scar pain, injury to palmar cutaneous branch of median nerve, incomplete recovery, re-innervation pain, incomplete release etc). When and how were symptoms measured? This section may need re-writing.
15. “The variant with the largest effect on CTS risk...” – how do you define largest effect? Is it by most statistically significant association? The Population attributable risk might be much less for this variant than others because it is such low frequency. I would remove this phrase, and just say the most statistically significant association.
16. I am not sure highlighting the SERPINA1 variant without some functional analysis adds much to the paper. I would recommend removing this section.

Discussion

17. Again, alter the recurrent CTS in the opening paragraph in light of point 13 above.
18. I would remove the sentence around SERPINA1 in the first paragraph.
19. Re-phrase the discussion paragraph around PRS in light of point 13 above.

Reviewer #2:

Remarks to the Author:

This is an extremely well written manuscript that is thorough and has novelty. I thoroughly enjoyed reading it. It adds significantly to the field, supporting previous genetic discoveries in CTS, expanding on them, and reporting new finds. It will stimulate new research. I could not see any obvious areas for improvement/change.

Response to reviewer's comments

We appreciate the time and effort the reviewers dedicated to improving the manuscript with their insightful and constructive comments. Most of the suggestions have been incorporated into the manuscript and Supplementary information using track changes. Below are our detailed responses.

Reviewer #2

This article describes a GWAS meta-analysis of Carpal Tunnel Syndrome. The authors discovered 53 variants at 50 loci, 32 of which have not previously been. I enjoyed reading the paper. The work is of high quality, and will be of interest to the scientific community. The paper overall is well-written. I have suggested changes to improve clarity of the manuscript, and I suggest acceptance after these comments are addressed.

- 1.1** I have not seen a data availability statement. It is vital that full summary statistics (not just the significant results presented in supplementary data) from these analyses are deposited in an appropriate public repository so they can be further used by the scientific community. This should be ensured by the editors prior to acceptance for publication.

Response (1.1)

The data availability statement (page 31) states that the GWAS summary statistics are available at <https://www.decode.com/summarydata/>.

Title

- 1.2** Title – replace associating with associated

Response (1.2)

We have replaced "...53 sequence variants associating with..." to "...50 genetic loci associated with..." according to reviewer's #1 comment and the editor's request.

Introduction

- 1.3** "contains the carpal bones, the transverse carpal ligament," – the carpal tunnel is bounded by these structures (they make up the anatomical boundary) it does not contain them.

Response (1.3)

We have corrected the phrasing from "contains" to "is bounded by" (page 3).

- 1.4** "The median nerve innervates muscles of the hand" change to "The median nerve innervates the thenar muscles and radial lumbricals" as the ulnar nerve innervates most of the intrinsic muscles of the hand.

Response (1.4)

We have reworded this sentence (page 3).

"The median nerve innervates the thenar muscles and the radial lumbricals..."

- 1.5** Diagnosis is based on a typical medical history in the first instance.

Response (1.5)

We have added that the medical history is important in the diagnostic process (page 4).

“The diagnosis of CTS is, together with medical history, obtained through medical assessment such as...”

- 1.6** remove “i.e. nerve conduction velocity across the carpal tunnel⁴” – electrophysiology is much more than conduction velocity.

Response (1.6)

We have removed “i.e. nerve conduction velocity across the carpal tunnel” from the text (page 4).

- 1.7** Treatment usually consists of lifestyle changes, night-time splintage, steroid injection, and surgical decompression.

Response (1.7)

The reviewer is correct. We mention lifestyle changes, steroid injection, and surgical decompression (page 4).

- 1.8** “sequence variants at 50 loci, thereof 32 novel.” Change to “sequence variants at 50 loci, of which 32 have not been previously reported”.

Response (1.8)

We have rephrased the sentence (page 4).

“...find 53 independent sequence variants at 50 loci, of which 32 have not been previously reported”

Methods

The methods are appropriate and comprehensive.

- 1.9** We estimated the genome-wide significance threshold using a weighted Bonferroni adjustment that controls for the family-wise error rate¹². Please refer to Supplementary table 2 here

Response (1.9)

We thank the reviewer for pointing this out. A reference to Supplementary Table 2 has been added (page 23).

Results

- 1.10** Figure 1 should be much larger on full submission, as it is very hard to interpret. Please highlight the previously unreported and the previously reported associations in a different colour from each other.

Response (1.10)

Figure 1 will be submitted as a separate file to ensure better quality. The previously reported and unreported variants have been highlighted with different colors.

1.11 Table 2 – please highlight in bold loci with previously undescribed associations

Response (1.11)

Table 2 was estimated to be too large for the main text by the editor. To address the editor’s request, we have removed Table 2 from the main text and refer to Supplementary Data 1 and 2 (page 5) where each sequence variant is marked with an ‘X’ if novel.

1.12 Table 2 – from which population is the reported EAF%? Please note in table legend

Response (1.12)

Table 2, with the combined effect allele frequency from the four populations, has been removed. In Supplementary Data 1 and 2, we list the combined effect allele frequency and the effect allele frequency in each population.

1.13 “the top 10% of pathogenic variants, thereof 10 in the top 1%” – change to “the top 10% of pathogenic variants, and 10 variants were in the top 1%”

Response (1.13)

We have simplified this sentence (page 10).

“...are among the top 10% of pathogenic variants, and 10 variants were in the top 1%”

1.14 PRS analysis: the definition of recurrent CTS is incorrect – this is a further operation not a recurrence. Could this be that people have had surgery on their left carpal tunnel and then >6months later had surgery on their right carpal tunnel? If you do not have data on side of surgery, recurrent should be limited to those with three or more surgical episodes (as no-one has three arms!), and the two or more surgical episodes (your group 3) should be redefined as either bilateral (most of these cases I think) or recurrent. The true recurrence rate in England is around 3% at 10 years (see figure 3 in doi: 10.1016/S2665-9913(20)30238-1). Please change this and Supplementary table 1. In the conclusion you state: “We did not have data on which hand was affected and could not define bilateral CTS cases specifically. Thus, we were not able to exclude the possibility of CTS in each hand at different timepoints in a case defined as recurrent.” This means it is indeed essential you change the nomenclature here. The vast majority of these cases will be bilateral, NOT recurrent.

Response (1.14)

We thank the reviewer for his expert opinion. As mentioned in the discussion, we did not have data on bilateral cases. To prevent losing power by only including those with three or more flare ups, we renamed group 3 as “bilateral, recurrent, or persistent”, that is, cases with two or more diagnosis with six months or more between health encounters (page 13 and Supplementary Table 1).

- 1.15** PRS: how did you define “symptoms after surgery”? The cause of such symptoms can be very broad (scar pain, injury to palmar cutaneous branch of median nerve, incomplete recovery, re-innervation pain, incomplete release etc). When and how were symptoms measured? This section may need re-writing.

Response (1.15)

We based all group definition on the number and date of ICD-10 diagnoses. The patients that underwent surgery and were diagnosed with CTS (ICD-10 code G56.0) after the surgery were considered as “symptoms after surgery”. We have changed the unspecific “CTS symptoms after surgery” to “CTS diagnosis after surgery”.

- 1.16** “The variant with the largest effect on CTS risk...” – how do you define largest effect? Is it by most statistically significant association? The Population attributable risk might be much less for this variant than others because it is such low frequency. I would remove this phrase, and just say the most statistically significant association.

Response (1.16)

We have replaced “with the largest effect on CTS risk” with “most significantly associated with CTS” (page 15).

- 1.17** I am not sure highlighting the *SERPINA1* variant without some functional analysis adds much to the paper. I would recommend removing this section.

Response (1.17)

We appreciate the reviewer’s suggestion. The most significant variant associating with CTS is the variant in *SERPINA1*. We find its protective effect on CTS particularly interesting, and worth highlighting, as it causes α -1 antitrypsin deficiency¹ and contributes to the risk of multiple traits, such as emphysema¹, chronic obstructive pulmonary disease¹, gallstone disease², and cirrhosis³. In addition, we show that the variant in *SERPINA1* affects 36 plasma proteins (page 15 and Supplementary Data 6).

Discussion

- 1.18** Again, alter the recurrent CTS in the opening paragraph in light of point 13 above.

Response (1.18)

We have modified “recurrent” to “bilateral, recurrent, or persistent” (page 16).

- 1.19** I would remove the sentence around *SERPINA1* in the first paragraph.

Response (1.19)

See Response (1.17)

- 1.20** Re-phrase the discussion paragraph around PRS in light of point 13 above.

Response (1.20)

We have rephrased the discussion paragraph on CTS PRS (page 18).

“Using CTS PRS, we found differences between nonrecurrent or nonpersistent and bilateral, recurrent, or persistent CTS cases, indicating higher genetic propensity for CTS among the latter group. We did not have data on which hand was affected and could not define bilateral CTS cases specifically. Thus, we combined cases that had two or more carpal tunnel flare ups with six or more months between healthcare encounters and compared them to cases with one carpal tunnel flare up.”

Reviewer #2

This is an extremely well written manuscript that is thorough and has novelty. I thoroughly enjoyed reading it. It adds significantly to the field, supporting previous genetic discoveries in CTS, expanding on them, and reporting new finds. It will stimulate new research.

I could not see any obvious areas for improvement/change.

References

1. Zorzetto, M. *et al.* SERPINA1 Gene Variants in Individuals from the General Population with Reduced α 1-Antitrypsin Concentrations. *Clin. Chem.* **54**, 1331–1338 (2008).
2. Ferkingstad, E. *et al.* Genome-wide association meta-analysis yields 20 loci associated with gallstone disease. *Nat. Commun.* **9**, 1–11 (2018).
3. Emdin, C. A. *et al.* Association of Genetic Variation With Cirrhosis: A Multi-Trait Genome-Wide Association and Gene–Environment Interaction Study. *Gastroenterology* **160**, 1620-1633.e13 (2021).